# Study on the Key Autotoxic Substances of Alfalfa and Their Effects

**DOI:** 10.3390/plants12183263

**Published:** 2023-09-14

**Authors:** Bei Wu, Shangli Shi, Huihui Zhang, Yuanyuan Du, Fang Jing

**Affiliations:** Key Laboratory of Grassland Ecosystem of Ministry of Education, College of Pratacultural Science, Gansu Agricultural University, Lanzhou 730070, China; wub@st.gsau.edu.cn (B.W.);

**Keywords:** alfalfa, phenolic autotoxic substances, coumarin, content, autotoxicity

## Abstract

Alfalfa is a leguminous plant with strong autotoxicity, which seriously affects regeneration and stability. In order to clarify the relationship between the key autotoxic substances and autotoxicity of alfalfa, this experiment determined the content of phenolic autotoxic substances in different varieties of alfalfa and the effect of different concentrations of alfalfa extracts on seed germination, seedling growth and physiology. The results showed that the content of single autotoxic substances in the eight alfalfa varieties was highest for total coumarin. The variety with the highest total coumarin content was “LZ”, and the lowest content was “656”. Principal component analysis of the autotoxicity of eight alfalfa varieties revealed that the variety with the strongest autotoxicity was “LZ” and the weakest was “656”. After treatment with extracts, the germination potential, germination rate, germination index and vigor index of 656 were higher than those of LZ, and the seeds of LZ and 656 did not germinate when the concentration was higher than C_0.025_ and C_0.05_, respectively. Compared with LZ, 656 had stronger osmotic regulation and antioxidant capacity, while the degree of membrane lipid peroxidation and ROS accumulation were lower. Further correlation analysis between the autotoxic substance content and autotoxicity observed that the content of total coumarin and autotoxic substances showed a significant positive association with autotoxicity (*p* < 0.01), and the total coumarin content showed a significant positive correlation with the content of autotoxic substances (*p* < 0.05). The total coumarin content is the major contributor to autotoxicity, and the higher the coumarin content, the higher the autotoxic substance content and the stronger the autotoxicity. Eight alfalfa varieties were systematically clustered on the basis of total coumarin content and autotoxicity, and the high-autotoxic alfalfa variety “LZ” and low-autotoxic alfalfa variety “656” were screened.

## 1. Introduction

Alfalfa (*Medicago sativa*) is an excellent perennial leguminosae which plays an important role in improving the ecological environment. It is a dominant grass species for the improvement and utilization of natural grasslands [1]. However, it has strong autotoxicity, which will lead to the problem of grassland renewal. As an important ecological chemical factor, autotoxicity has an impact on material circulation, energy flow, system succession and system diversity in the ecosystem, which seriously affects the renewal and stability of the ecological community of alfalfa grassland and will cause the cliff-like decline of alfalfa ecological grassland [2]. Allelopathy refers to the phenomenon that contemporary donor plants synthesize different kinds and concentrations of secondary metabolites, which are released into the ecological environment through volatilization, leaching, root secretion or residue decomposition, directly or indirectly inhibiting their growth and that of surrounding plants. Autotoxicity is an intraspecific manifestation of allelopathy in species [3,4]. The effects of autotoxicity are mainly reflected in the changes in the morphological structure, physiological function, material metabolism and gene expression of seeds and seedlings. Autotoxic substances are the secondary metabolites and their derivatives cause autotoxicity in plants [5,6]. Therefore, it is particularly important to explore the key autotoxic substances in alfalfa and their relationship to autotoxicity.

In recent years, many different types of autotoxic substances have been isolated and identified from plants, including phenolic compounds (flavonoids, phenolic acids, coumarins, tannins and cinnamic acid derivatives, etc.), terpenoids (terpenes, alcohols, aldehydes, ketones and sesquiterpenes, etc.) and nitrogenous organic compounds (alkaloids, glucosinolates and cyanogenic glycosides, etc.) [7,8]. The phenolic substances in wheat (*Triticum aestivum* L.) are benzoic acid, vanillic acid, coumarin, ferulic acid, cinnamic acid, etc. [9]. Autotoxic substances such as ferulic acid, cinnamic acid and coumarin are found in the rhizosphere soil of *Paeonia ostii* T. Hong et J. X. [10]. Studies of the autotoxic substances in alfalfa have indicated that alfalfa leaves contain salicylic acid, p-hydroxybenzoic acid, scopoletin, quercetin and other autotoxic substances [11]. Rong detected the autotoxic substances in root exudates and residues of different alfalfa varieties, and obtained coumarin, coumaric acid, p-hydroxybenzoic acid, caffeic acid, chlorogenic acid and ferulic acid [6]. Lu et al. also detected four phenolic compounds, coumarin, o-coumaric acid, m-coumaric acid and p-coumaric acid, in the extract of alfalfa leaves [12].

With the isolation and identification of various autotoxic substances from plants and soils, more and more studies have begun to focus on exploring the mechanism of plant autotoxicity. There are many studies have shown that autotoxicity significantly inhibits seed germination, seedling phenotypic shaping, physiological and biochemical metabolism and gene expression in their offspring [13,14]. Phenotypic changes were reflected in the decrease in seed germination rate, germination potential, germination index, the inhibition of seed vigor, and the inhibition of radicle and germ elongation. The autotoxicity inhibition was enhanced as the autotoxic substance content increased [15]. Long et al. found that seed germination potential and germination rate could be significantly inhibited by the water extract of *Polygonatum cyrtonema* plants and rhizosphere soil, and with the increase in stress degree, the autotoxicity inhibition was enhanced [16]. With increasing extract concentration, seed germination potential, germination rate, germination index and vigor index decreased significantly, seedling radicle length, germ length, biomass and root activity decreased significantly [6]. Autotoxic stress can also lead to changes in the physiological and biochemical responses of offspring seedlings, such as the induction of reactive oxygen species (ROS) production and gene expression. High levels of ROS cause oxidative stress, alter cell membrane permeability and exacerbate membrane lipid peroxidation [17]. ROS homeostasis is strictly controlled by the plant antioxidant defense system. The defense enzymes such as superoxide dismutase (SOD), peroxidase (POD) and catalase (CAT) interact and coordinate with each other in the enzymatic reaction system to reduce the damage of ROS to plants [18,19]. In addition to the antioxidant defense system, the accumulation of osmotic adjustment substances such as soluble sugar (SS), soluble protein (SP) and proline (Pro) can also protect the cell structure to resist the damage caused by stress, which is one of the main ways to maintain intracellular homeostasis [20]. It has been reported that roots respond to stress by activating antioxidant enzymes and increasing the content of osmotic adjustment substances after 12 h of autotoxicity treatment in melon. The activities of SOD and CAT increased rapidly, and the contents of SS, SP and Pro increased significantly under autotoxicity [21]. The phenolic autotoxic substances in alfalfa have an effect on the activity of SOD and POD in receptor plants, thus disturbing the balance of active oxygen metabolism in the receptor, causing membrane damage and a significant increase in malonaldehyde (MDA) levels [22]. In addition, alfalfa can be divided into autotoxicity tolerant and autotoxicity sensitive varieties. There are differences between them in terms of phenotypic and physiological responses to autotoxic stress. Autotoxicity-tolerant alfalfa has greater antioxidant capacity and osmotic adjustment ability than autotoxicity sensitive alfalfa [23].

Previous autotoxicity studies have mainly focused on the isolation and identification of plant autotoxic substances and the effects of autotoxic stress on seed germination and seedling growth. However, the key autotoxic substances of alfalfa, the differences in the response of different alfalfa varieties to autotoxic stress, and the relationship between the synthesis and accumulation of autotoxic substances and autotoxicity are still unclear. This study aims to: (1) determine the content of phenolic autotoxic substances in alfalfa, and which substance is mainly involved in the autotoxic stress inhibition of seed germination and seedling growth; (2) elucidate differences in the response of alfalfa varieties to autotoxic stress in terms of germination, seedling morphology and physiological response (ROS production, osmotic regulation, antioxidant capacity and gene expression); (3) clarify the relationship between the content of autotoxic substances and the autotoxicity, and the contribution of each autotoxic substance to the autotoxicity. Screening of high- and low-autotoxic alfalfa varieties with different contents of autotoxic substance and autotoxicity. It provides a theoretical basis for further exploring the anabolism mechanism of autotoxic substances and creating low-autotoxic alfalfa varieties.

## 2. Results

### 2.1. Differences in the Content of Phenolic Autotoxic Substances in Alfalfa

The content of individual phenolic autotoxic substances in the eight alfalfa varieties showed the highest total coumarin content, followed by ferulic acid. Vanillic acid and p-hydroxybenzoic acid were lower (Figure 1). The variety with the highest total coumarin content was LZ, which was 28.60 mg·g^−1^, and the lowest content was 656, which was 30.62 mg·g^−1^. LZ was significantly higher than 656 by 7.06% (*p* < 0.05). The variety with the highest content of ferulic acid was LZ, and the lowest was 4114. The content of vanillic acid was the highest in 3105, which was 25.83 μg·g^−1^, and the content of 656 and X4 was lower, which was 15.71 μg·g^−1^ and 15.43 μg·g^−1^, respectively. 3105 was significantly higher than 656 and X4 by 64.41% and 67.41%, respectively (*p* < 0.05). The content of p-hydroxybenzoic acid was the highest in QS and the lowest in G9. In addition, the content of main phenolic autotoxic substances in leaves of eight alfalfa varieties ranged from 29.39 mg·g^−1^ to 31.56 mg·g^−1^, the content of LZ was the highest, and the content of 656 was the lowest, LZ was significantly higher than 656 by 7.38% (*p* < 0.05).

### 2.2. Effect of Alfalfa Extracts on Seed Germination and Seedling Growth

#### 2.2.1. Effect of Alfalfa Extracts on Seed Germination

Germination potential

With increasing extract concentration, the germination potential of 656, 4114, 4010 and 3105 seeds tended to increase and then decrease (Figure 2A). The germination potential of 656 was the largest at C_0.001_, and the inhibition effect of the extract reached a significant level at C_0.025_, which was significantly lower than that of the control by 0.93% (*p* < 0.05). The germination potential of 4114, 4010 and 3105 was the highest at C_0.005_. At C_0.025_, the inhibitory effect of the extract reached a significant level, which was significantly lower than that of the control by 51.43%, 36.37% and 100.00%, respectively (*p* < 0.05). G9, LZ and QS extracts were found to inhibit seed germination potential in a concentration gradient, with seed germination potential decreasing gradually. The inhibition of G9 and LZ reached a significant level at C_0.025_, which was significantly lower than that of the control by 5.45% and 12.50%, respectively (*p* < 0.05). QS was significantly lower than that of the control by 4.81% at C_0.001_ (*p* < 0.05). Under the same concentration treatment, the germination potential of LZ was significantly lower than that of G9, 656 and 4114 (*p* < 0.05).

Germination rate

With increasing concentration of the extract, the seed germination rate of 656, LZ, 4114, 4010 and 3105 showed a tendency first to increase and then to decrease (Figure 2B). The germination rate of 656 and 3105 was the highest at C_0.001_, LZ, 4114 and 4010 was the highest at C_0.005_. The inhibitory effect of 656 reached a significant level at C_0.01_, which was significantly lower than that of the control by 2.72% (*p* < 0.05). The inhibitory effects of LZ, 4114 and 3105 reached a significant level at C_0.025_. As the extract concentration increased, the seed germination rate of G9, X4 and QS gradually decreased. The inhibition of QS and G9 reached a significant level at C_0.005_, which was significantly lower than that of the control by 9.01% and 13.40% (*p* < 0.05). Under the same concentration treatment, the germination rate of 656 was significantly higher than that of LZ (*p* < 0.05).

Germination index

With increasing concentration of the extract, the seed germination index of 656, LZ, 4114, 4010 and 3105 showed a tendency first to increase and then to decrease (Figure 2C). The germination index was the largest at C_0.001_, and the inhibition effect reached a significant level at C_0.01_, which was significantly lower than that of the control by 16.20% and 21.94% (*p* < 0.05). With the increase in extract concentration, the germination index of G9, X4, LZ, 4010, QS and 3105 decreased gradually. The inhibitory effect of G9 and LZ extracts at C_0.005_ reached a significant level, which was significantly lower than that of the control by 17.59% and 11.47%, respectively (*p* < 0.05). The germination index of X4, QS and 3105 at C_0.001_ was significantly lower than that of the control by 8.63%, 10.78% and 7.27%, respectively (*p* < 0.05). At C_0.01_, the germination index of 4010 was significantly lower than that of the control by 20.61% (*p* < 0.05).

Vital index

With increasing concentration of the extract, the seed vital index of 656, LZ, 4114, 4010 and 3105 showed a tendency first to increase and then to decrease (Figure 2D). G9, 656 and QS had the highest activity index at C_0.001_, G9 and 656 reached a significant level, which were significantly higher than the control by 34.53% and 29.12% (*p* < 0.05). With the increase in extract concentration, the vital index of X4 and 3105 decreased gradually, and the inhibitory effect gradually increased. The inhibitory effect reached a significant level at C_0.001_, which was significantly lower than that of the control by 29.22% and 15.87%, respectively (*p* < 0.05). Under the same concentration treatment, the vital index of LZ was the lowest, which was significantly lower than that of G9, 656, 4114 and 4010 (*p* < 0.05), and the vital index of 656 was significantly higher than that of QS (*p* < 0.05).

#### 2.2.2. Effect of Alfalfa Extracts on Seedling Growth

Root length

With increasing extract concentration, the root length of the eight varieties of alfalfa seedlings tended to first increase and then decrease (Figure 3A). 656 and X4 were the longest at C_0.001_ and C_0.005_, respectively, and the other varieties were the longest at C_0.01_. At C_0.025_, the inhibitory effects of X4, LZ, 4010, QS and 3105 reached a significant level, compared with the control (*p* < 0.05). At C_0.05_, the inhibitory effects of G9, 656 and 4114 reached a significant level, which were significantly reduced by 100.00% compared with the control (*p* < 0.05). Compared with other varieties, the inhibitory effect of G9 and 4010 was lower, and the inhibitory effect of LZ was the highest. The root length of QS was significantly lower than that of G9 and 4010 (*p* < 0.05), and LZ was significantly lower than 656 (*p* < 0.05).

Seedling length

With increasing concentration of the extract, the seedling length of G9, 656, LZ, 4114, 4010 and QS showed a trend of increasing first and then decreasing (Figure 3B). G9, 656 and 4010 were the longest at C_0.001_, and the promotion effect reached a significant level, which was significantly higher than that of the control by 55.22%, 31.69% and 78.63% (*p* < 0.05), and the promotion effect changed from promotion to significant inhibition at C_0.01_, C_0.005_ and C_0.025_, respectively. With the increase in extract concentration, the seedling length of X4 and 3105 decreased gradually, and significantly decreased by 30.72% and 12.64% compared with the control at C_0.001_ (*p* < 0.05). At the same concentration, the seedling length of LZ was significantly different from that of G9, 656 and 4114. At C_0.005_ and C_0.01_, LZ was significantly higher than G9 and 656 (*p* < 0.05), and at C_0.05_ and C_0.025_, LZ was significantly lower than G9, 656 and 4114 (*p* < 0.05).

Fresh weight

With increasing concentration of the extract, the fresh weight of the eight varieties of alfalfa seedlings showed a tendency first to increase and then to decrease (Figure 3C). X4 and 4010 were the highest at C_0.001_, and the promoting effect of the extract was the largest, which changed from a promoting effect to an inhibiting effect at C_0.005_ and C_0.01_, respectively. The fresh weight of G9, 4114 and 3105 was the highest at C_0.005_. At C_0.05_, G9 was significantly reduced by 100.00% compared with the control (*p* < 0.05). At C_0.025_, 4114 and 3105 changed from promotion to inhibition, and the inhibition of 3105 reached a significant level (*p* < 0.05). The fresh weight of 656, LZ and QS was the highest at C_0.01_, which was significantly higher than that of the control by 19.53%, 57.52% and 23.42%, respectively (*p* < 0.05). At the same concentration, the fresh weight of LZ was the lowest, which was significantly lower than that of G9, 656, 4114 and 4010 (*p* < 0.05).

Dry weight

With increasing concentration of the extract, the dry weight of the eight varieties of alfalfa seedlings showed a tendency first to increase and then to decrease (Figure 3D). X4 and 4010 were the highest at C_0.001_, and the promotion effect of X4 reached a significant level, which was significantly increased by 25.00% compared with the control (*p* < 0.05). At C_0.025_ and C_0.005_, the promoting effect of X4 and 4010 changed from promotion to significant inhibition, which was significantly reduced by 100.00% and 7.69%, respectively, compared to the control (*p* < 0.05). The dry weight of LZ and 4114 was the highest at C_0.005_, which was significantly higher than that of the control by 40.48% and 12.12% (*p* < 0.05). At C_0.025_ and C_0.01_, the dry weight of LZ and 4114 changed from promoting to significantly inhibiting, which was significantly lower than that of the control by 100.00% and 6.06% (*p* < 0.05). The dry weight of 656, QS and 3105 was the largest at C_0.01_, 656 changed from promotion to inhibition at C_0.05_, and QS and 3105 changed from promotion to inhibition at C_0.025_. At the same concentration, the dry weight of QS was significantly lower than that of 4010 (*p* < 0.05).

#### 2.2.3. The Autotoxicity of Alfalfa Extracts on Seed Germination and Seedling Growth

Table 1 shows the allelopathic effect index of different alfalfa extract concentrations on GP, GR, GI, VI, RL, SL, FW and DW.

RI (GP)

The RI (GP) values of G9, X4, LZ and QS were less than 0 when treated with each extract concentration and the inhibitory effect gradually increased with increasing concentration. Both 656 and 4114 changed from promotion to inhibition at C_0.01_. At C_0.025_, 4010 and 3105 changed from a promoting effect to an inhibiting effect; with increasing concentration, the promoting effect first increased and then decreased, and the inhibiting effect increased gradually.

RI (GR)

For each extract concentration treatment, the RI (GR) values of G9, X4 and QS were all less than 0, and the inhibition effect gradually increased with increasing concentration. Both 4114 and 3105 changed from a promoting effect to an inhibiting effect at C_0.025_. With increasing concentration, the promoting effect of 3105 gradually decreased, while LZ and 4114 increased first and then decreased. 656 and 4010 changed from promotion to inhibition at C_0.01_ and C_0.05_, respectively.

RI (GI)

The RI (GI) values of G9, X4, 4010, QS and 3105 remained below 0 for each extract concentration and the inhibitory effect gradually increased with increasing concentration. At C_0.005_, 656, LZ and 4114 changed from a promoting effect to an inhibiting effect.

RI (VI)

The RI (VI) values of X4 and 3105 were less than 0 under the treatment of each concentration of extract, 4114 and 4010 were less than 0 when they were higher than C_0.01_. G9, 656 and QS changed from promotion to inhibition at C_0.005_. The RI (VI) value of LZ was greater than 0 when it was lower than C_0.025_, and it became an inhibitory effect at C_0.025_.

RI (RL)

Under the treatment of each extract concentration, the RI (RL) values of X4 and 3105 were less than 0, and the inhibition gradually increased with increasing concentration. 656 changed from a promoting effect to an inhibiting effect at C_0.005_. When the concentration of the extract was lower than C_0.025_, the RI (RL) values of LZ, 4114, 4010 and QS were all greater than 0. With increasing extract concentration, the promoting effect of 4010 gradually increased, the other varieties increased first and then decreased, and the inhibiting effect gradually increased.

RI (SL)

G9 and 4010 changed from promotion to inhibition at C_0.05_. At C_0.025_, X4, LZ, 4114, QS and 3105 switched from promotion to inhibition. At C_0.005_, 656 changed from a promoting effect to an inhibiting effect. The inhibiting effect gradually increased with increasing concentration.

RI (FW)

G9 changed from a promoting effect to an inhibiting effect at C_0.05_. X4 and 4010 changed from a promoting effect to an inhibiting effect at C_0.01_, the promoting effect gradually decreased and the inhibiting effect gradually increased with increasing concentration. 656, LZ, 4114, QS and 3105 changed from a promoting effect to an inhibiting effect at the concentration of C_0.025_.

RI (DW)

G9 and 656 changed from a promoting effect to an inhibiting effect at C_0.05_. With the increase in extract concentration, the promoting effect of G9 increased gradually, and the promoting effect of 656 increased first and then decreased. With increasing extract concentration, the promoting effect of QS and 3105 increased gradually, the promoting effect of X4 decreased gradually, and the promoting effect of LZ increased first and then decreased. 4114 changed from a promoting effect to an inhibiting effect at C_0.01_.

### 2.3. Effect of Alfalfa Extracts on Seedling Physiology

#### 2.3.1. Effect of Alfalfa Extracts on Antioxidant Enzyme Activity of Seedlings

POD activity

With increasing extraction concentration, the POD activity of all alfalfa varieties tends to increase and then decrease (Figure 4A); and except for 4010 and QS, the POD activity of other varieties was significantly higher than that of the control (*p* < 0.05). The activities of G9, LZ, 4010, QS and 3105 were the highest at C_0.001_, which were significantly higher than those of the control by 18.94%, 32.20%, 13.54% and 75.60% (*p* < 0.05). The POD activities of 656, X4 and 4114 were the highest at C_0.005_, which were significantly increased by 114.13%, 103.80% and 39.61%, respectively, compared with the control (*p* < 0.05). Under the same concentration treatment, the POD activity of LZ was significantly lower than that of G9, 656, X4, 4114 and 4010 (*p* < 0.05).

SOD activity

With increasing extract concentration, the SOD activity of G9 increased gradually, and it was significantly increased by 59.86% compared with the control at C_0.01_ (*p* < 0.05) (Figure 4B). X4, LZ and 4114 showed a trend of increasing first and then decreasing with increasing concentration, the highest at C_0.001_, which was significantly higher than that of the control by 4.61%, 12.31% and 11.18%, and the SOD activity was significantly lower than that of the control when higher than C_0.001_ (*p* < 0.05). The SOD activity of 656 was significantly higher than that of the control, and the activity was the highest at C_0.01_ (*p* < 0.05). The SOD activity of 4010, QS and 3105 decreased first and then increased with the increase in concentration, reaching the lowest at C_0.001_, C_0.005_ and C_0.005_, respectively. Under the same concentration treatment, the SOD activity of G9 was significantly lower than that of 656, LZ, 4114, 4010 and 3105 (*p* < 0.05). When the concentration was lower than C_0.01_, the SOD activity of 656 was significantly lower than that of LZ, 4114, 4010 and 3105, and 656 was significantly higher than LZ and 4114 at C_0.01_ (*p* < 0.05).

CAT activity

With the increase in extract concentration, the CAT activity of G9, LZ, 4114, 4010, QS and 3105 increased first and then decreased, and the CAT activity of 656 increased gradually (Figure 4C). The CAT activities of G9, 656, 4010, QS and 3105 were significantly higher than those of the control. G9 and 656 were the highest at C_0.005_ and C_0.01_. The CAT activity of 4010, QS and 3105 was the highest at C_0.001_, which was significantly higher than that of the control by 103.89%, 27.62% and 38.64%, respectively (*p* < 0.05). The CAT activity of LZ and 4114 was the lowest at C_0.005_ and C_0.01_. The activity of X4 was the lowest at C_0.001_, which was significantly lower than that of the control by 20.56% (*p* < 0.05). In addition, the CAT activity of 656 was significantly lower than that of X4, LZ, 4010, QS and 3105 (*p* < 0.05).

#### 2.3.2. Effect of Alfalfa Extracts on the Content of Osmotic Adjustment Substances in Seedlings

SS content

With increasing extract concentration, the SS content of G9, 656, LZ, 4114, 4010 and 3105 showed a trend of increase–decrease–increase, but they were significantly higher than that of the control (Figure 5A). G9, 656, 4114 and 4010 were the highest at C_0.01_, which was significantly increased by 99.65%, 56.21%, 80.34% and 81.72%, respectively, compared with the control (*p* < 0.05). The SS content of LZ and 3105 was the highest at C_0.001_, which was significantly higher than that of the control by 59.60% and 72.26%, respectively (*p* < 0.05). With the increase in concentration, the SS content of X4 increased first and then decreased, and the SS content of QS increased gradually. The level of X4 and QS were highest at concentrations of C_0.005_ and C_0.01_, respectively, which were significantly increased by 53.67% and 93.01% compared with the control (*p* < 0.05). Under the same concentration treatment, the SS content of 656 was higher than that of G9, LZ, 4114, 4010 and QS, and there were significant differences between 656 and G9, 4114 and 4010.

SP content

With the increase in extract concentration, the SP content of G9, X4, LZ and 4010 increased first and then decreased (Figure 5B). The content of X4 was the highest at C_0.005_, which was significantly higher than that of the control by 44.82% (*p* < 0.05). The content of G9, LZ and 4010 was the highest at C_0.001_, which was significantly higher than that of the control by 30.76%, 10.97% and 19.47% (*p* < 0.05). With increasing concentration, the SP content of 4114 decreased gradually, and the SP content of 656, QS and 3105 decreased first and then increased. In addition, under the same concentration treatment, the SP content of G9 was significantly lower than 4114 and 4010, X4 was significantly lower than 4010 (*p* < 0.05).

Pro content

Compared with the control, the Pro content of eight alfalfa varieties was significantly higher than that of the control (*p* < 0.05). With the increase in concentration, the Pro content of QS and 3105 increased gradually, and the Pro content of the other six varieties showed a trend of increase–decrease–increase (Figure 5C). In addition, G9 had the highest Pro content at C_0.001_, and other varieties had the highest content at C_0.01_. Compared with the control, LZ had the largest increase, which was significantly increased by 135.91% (*p* < 0.05).

#### 2.3.3. Effect of Alfalfa Extracts on MDA and H_2_O_2_ Content in Seedlings

The contents of MDA and H_2_O_2_ in eight alfalfa varieties were significantly higher than in the control (*p* < 0.05). With increasing concentration, the content of MDA and H_2_O_2_ in G9 and QS increased gradually, and the content of MDA and H_2_O_2_ in 656, X4, LZ, 4114, 4010 and 3105 increased first and then decreased.

MDA content

The MDA content of 656 was highest at C_0.001_, which was significantly higher than that of the control by 42.73% (*p* < 0.05). X4, LZ, 4114, 4010 and 3105 was highest at C_0.005_, which was significantly higher than that of the control by 27.11%, 18.15%, 33.13%, 42.38% and 53.77%, respectively (*p* < 0.05). At C_0.01_, the MDA content of G9 and QS was highest, significantly higher than the control at 31.77% and 32.01%, respectively (*p* < 0.05). Under the same concentration treatment, the MDA content of LZ was significantly higher than that of G9, 656, X4, 4114,4010 and 3105 (*p* < 0.05) (Figure 6A).

H_2_O_2_ content

The H_2_O_2_ content of 656 and 3105 was highest at C_0.001_, which was significantly higher than that of the control by 14.68% and 5.71%, respectively (*p* < 0.05). X4, LZ, 4114 and 4010 was highest at C_0.005_, which was significantly higher than that of the control by 11.83%, 17.38%, 19.20% and 3.03%, respectively (*p* < 0.05). The H_2_O_2_ content of G9 and QS was the highest at C_0.01_, which was significantly higher than that of the control by 12.97% and 7.41%, respectively (*p* < 0.05) (Figure 6B).

#### 2.3.4. The Autotoxicity of Alfalfa Extracts on Seedling Physiology

Table 2 shows the allelopathic effect index of different alfalfa extract concentrations on POD, SOD, CAT, SS, SP, Pro, MDA and H_2_O_2_.

RI (POD)

With increasing extract concentration, the promoting effect of G9, LZ, 4010, QS and 3105 gradually decreased, the promoting effect of 4010 and QS changed from promoting to inhibiting at C_0.01_ and C_0.005_, respectively. The RI (POD) values of 656, X4 and 4114 were all greater than 0, and with increasing extract concentration, the promoting effect showed a trend of increasing first and then decreasing.

RI (SOD)

X4, LZ and 4114 changed from promotion to inhibition at C_0.005_, and the inhibition of X4 and 4114 gradually increased with increasing extract concentration. The RI_SOD_ values of G9 and 656 were greater than 0. With the increase in extract concentration, the promoting effect of G9 increased first and then decreased, 656 decreased first and then increased. The RI (SOD) value of 3105 was less than 0, and the inhibitory effect gradually increased with increasing extract concentration. 4010 and QS changed from inhibition to promotion at C_0.001_ and C_0.005_, respectively.

RI (CAT)

When treated with different extract concentrations, the RI (CAT) values of G9, 656, 4010, QS and 3105 were greater than 0. With increasing concentration, the promoting effect of G9 increased first and then decreased, the promoting effect of 4010 decreased first and then increased, and the promoting effect reached the maximum at C_0.005_ and C_0.001_, respectively. As the extract concentration increased, the promoting effect of 656 gradually increased, and the promoting effect of QS and 3105 gradually decreased. X4 changed from inhibition to promotion at C_0.005_. 4114 changed from promotion to inhibition at C_0.005_, and the inhibition gradually increased with increasing concentration.

RI (SS)

The RI (SS) values of eight alfalfa varieties were greater than 0 at each concentration. With increasing extract concentration, the promoting effect of G9, 656, LZ, 4114, 4010 and 3105 decreased first and then increased, being lowest at C_0.005_. The promoting effect of G9, 656, 4114 and 4010 reached its maximum at C_0.01_, LZ and 3105 at C_0.001_. The promoting effect of QS gradually increased. The promoting effect of X4 increased first and then decreased, reaching a maximum at C_0.005_.

RI (SP)

Under the treatment of each extract concentration, the RI (SP) values of G9 and X4 were greater than 0. The promoting effect of G9 decreased first and then increased, X4 increased first and then decreased. After treatment with the extract, the RI (SP) values of 4114, QS and 3105 were all less than 0. The inhibitory effect of 4114 gradually increased, while QS and 3105 increased first and then decreased. LZ changed from a promoting effect to an inhibiting effect at C_0.005_. With increasing extract concentration, the inhibitory effect of 656 decreased gradually, changing to a promotion effect at C_0.01_. 4010 changed from a promoting effect to an inhibiting effect at C_0.01_.

RI (Pro)

The RI (Pro) values of 656, X4, LZ, 4114, 4010 and 3105 were all less than 0 under the treatment of different concentrations of extracts. With increasing extract concentration, the inhibitory effect of 3105 increased first and then decreased, and the other varieties increased gradually. At C_0.005_, G9 changed from a promoting effect to an inhibiting effect. QS changed from inhibition to promotion at C_0.01_.

RI (MDA)

Under the treatment of different extract concentrations, the RI (MDA) values of eight alfalfa varieties were all greater than 0. With increasing concentration, the promoting effect of G9 and QS increased gradually, 656 decreased gradually, X4, LZ, 4114, 4010 and 3105 increased first and then decreased.

RI (H_2_O_2_)

Under the treatment of different extract concentrations, the RI (H_2_O_2_) values of eight alfalfa varieties were all greater than 0. With increasing concentration, the promoting effect of G9 and QS increased gradually, 656 and 3105 decreased gradually, X4, LZ, 4114 and 4010 increased first and then decreased.

### 2.4. Comprehensive Analysis of Alfalfa Autotoxicity

Principal component analysis was performed on 16 indexes of GP, GR, GI, VI, RL, SL, FW, DW, POD, SOD, CAT, SS, SP, Pro, MDA and H_2_O_2_ of eight alfalfa varieties. Six principal components were extracted and the cumulative contribution rate reached 86.61% (Figure 7). According to the principles of KMO > 0.5 and Bartlett = 0.000, initial eigenvalue > 1 and cumulative variance contribution rate > 85%, these six principal components can be considered to represent most of the information of alfalfa autotoxicity. The contribution rate of the first principal component was 32.22%. The positive load weights of GP, GR, GI and VI were larger than 0.80, which had a greater positive impact on the first principal component. The negative load weights of MDA and H_2_O_2_ were larger than 0.60, which had a greater negative impact on the first principal component. The contribution rate of the second principal component is 17.26%, and the positive load weight of SS is the largest, which is 0.72. The contribution rate of the third principal component is 11.54%, and the SP was main index and the load weight was 0.88. The contribution rate of the fourth principal component is 9.91%, SL and CAT were the positive and negative index, respectively, and the load weights were 0.72 and 0.52. The contribution rate of the fifth principal component is 8.71%, and the index with larger positive load value is SL, which is 0.57. The contribution rate of the sixth principal component is 6.97%, and the largest index of positive load value is POD, which is 0.53. In summary, GP, GR, GI, VI, MDA, H_2_O_2_, SS, SP, RL, CAT and POD can be used to effectively evaluate indexes for the autotoxicity of different alfalfa varieties.

The principal component comprehensive model is calculated by using the ratio of the six principal components and the eigenvalues corresponding to each principal component to the sum of the eigenvalues of all the extracted principal components as the weights:Y_comprehensive_ = 0.32223 × Y_1_ + 0.17257 × Y_2_ + 0.11537 × Y_3_ + 0.09907 × Y_4_ + 0.08711 × Y_5_ + 0.06972 × Y_6_(1)

The results showed that the comprehensive scores of G9 were 0.55, 0.18 and 0.49 under the treatment of C_0.001_, C_0.005_ and C_0.01_ extracts, respectively. The comprehensive scores of 656 were 1.15, 0.83, 1.31. The comprehensive scores of X4 were 0.40, 0.63, −0.40, respectively. The comprehensive scores of LZ were −1.08, −1.54, −1.52. The comprehensive scores of 4114 were 0.80, 0.79, 0.39, respectively. The comprehensive scores of 4010 were 0.21, 0.22, −0.15, respectively. The comprehensive scores of QS were −0.88, −1.11, −0.89, respectively. The comprehensive scores of 3105 were 0.85, 0.43, 0.63, respectively (Figure 8A). In addition, the comprehensive scores of eight alfalfa autotoxicity were ranked, with LZ having the strongest autotoxicity and 656 having the weakest autotoxicity (Figure 8B).

Finally, the seed germination and seedling growth of high-autotoxic alfalfa LZ and low-autotoxic alfalfa 656 after extract treatment were compared. The seeds of LZ did not germinate when the extract concentration was higher than C_0.025_, and the seeds of 656 did not germinate when the extract concentration was higher than C_0.05_ (Figure 9A,B).

### 2.5. Study on the Relationship between the Content of Phenolic Autotoxic Substances and Autotoxicity in Alfalfa

#### 2.5.1. Correlation Analysis between the Content of Phenolic Autotoxic Substances and Autotoxicity in Different Alfalfa Varieties

The correlation between the content of phenolic autotoxic substances in different varieties of alfalfa and the comprehensive score of autotoxicity was analyzed, and the correlation coefficient matrix was obtained (Figure 10). The autotoxicity of different alfalfa varieties was significantly positively correlated with the content of total coumarin and autotoxic substances (*p* < 0.01), and the correlation coefficients were 0.86 and 0.89, respectively. Among them, there was a significant positive correlation between the content of autotoxic substances and the content of total coumarin (*p* < 0.01), and the correlation coefficient was 1.00. In addition, there was a significant positive correlation between vanillic acid content and caffeic acid content (*p* < 0.05), and the correlation coefficient was 0.73.

#### 2.5.2. Analysis of the Contribution of Alfalfa Single Autotoxic Substances to Autotoxicity

In this experiment, the content of phenolic autotoxic substances in different varieties of alfalfa was determined, and the contribution rate of a single autotoxic substance was calculated (Figure 11). The content of phenolic autotoxic substances in eight alfalfa varieties showed that the contribution rate of total coumarin was the largest, higher than 96.00%. The contribution rate of p-hydroxybenzoic acid and vanillic acid was low, less than 0.10%.

### 2.6. Cluster Analysis of Alfalfa Varieties

In this paper, based on the comprehensive score of autotoxicity and total coumarin content, eight alfalfa varieties were systematically clustered by Euclidean distance class average method (Figure 12). The results showed that the alfalfa could be divided into three different types at the level of Euclidean distance *d* = 10. There were two varieties in the first type, QS and LZ. It was a highly autotoxic variety with high total coumarin content and autotoxicity. There was one variety (656) in the second category. This type of alfalfa had the lowest total coumarin content and autotoxicity, which was a low-autotoxic variety. There were five varieties in the third category, which were 4010, X4, 4114, 3105 and G9, respectively. The alfalfa total coumarin content and autotoxicity effect of this type were at a medium level, which was a medium autotoxic variety.

### 2.7. Construction of Autotoxicity Mechanism Model of Alfalfa

In order to better understand the process of alfalfa from producing autotoxic substances to autotoxicity. The key indexes of phenolic autotoxic substances, seed germination, seedling phenotype, defense enzymes, osmotic regulators and oxidative damage of the high-autotoxic variety “LZ” and the low-autotoxic variety “656” selected in this paper were used to construct a physiological metabolic model (Figure 13). In general, the process of autotoxicity can be divided into three stages: (1) The synthesis of autotoxic substances from donor plants. (2) The release of autotoxic substances. (3) Autotoxicity to recipient plants. The contents of total coumarin in LZ and 656 were the highest, and the contents of vanillic acid and p-hydroxybenzoic acid were lower. Coumarin played a key role in the process of producing autotoxicity. Compared with LZ, 656 synthesized a lower content of total coumarin, which was released into the air and soil by volatilization, leaching, root secretion and residue decomposition, and had a weaker autotoxicity on seed germination and seedling growth of offspring. The germination rate, germination potential, germination index and vigor index of 656 were higher than those of LZ. The membrane lipid peroxidation of 656 was low, and it could effectively scavenge reactive oxygen species (H_2_O_2_). By promoting the accumulation of osmotic adjustment substances (SS, SP, Pro) and the increase in antioxidant enzyme activities (POD, SOD and CAT), the autotoxicity of 656 was lower than that of LZ.

## 3. Discussion

### 3.1. The Response of Alfalfa Seed Germination, Seedling Growth and Seedling Physiology to Extracts Treatment

#### 3.1.1. The Response of Alfalfa Seed Germination and Seedling Growth to Extracts Treatment

The secondary metabolites and their derivatives that cause plant autotoxicity are autotoxic substances, which are not a single substance, but a mixture of several substances [24]. Therefore, when exploring the autotoxicity of plants, the method of water extract preparation is usually used to extract autotoxic substances (Neumann et al., 2009), which can eliminate the influence of external environment and reflect the essence of the extract to the greatest extent [25]. Seed germination is the primary stage of plant growth, which is sensitive to changes in the external environment [26,27,28], and may directly reflect plant autotoxicity [29]. Previous studies have reported that the autotoxicity on seed germination was closely related to plant varieties. Tibugari et al. found that the germination of three varieties of sorghum seeds was inhibited under the treatment of sorgoleone, but the germination inhibition rate of different varieties was different [30]. Similarly, there were significant differences in the inhibition rate of seed germination and radicle growth of different wheat water extracts [31]. In addition, the autotoxicity of stem and root extracts from three different safflower varieties on seed germination (germination rate) and seedling growth (rhizome length, fresh weight and dry weight) differed significantly [32]. In this study, it was found that the effects of different alfalfa extracts on seed germination rate, germination potential, germination index and vigor index were significantly different. By calculating the allelopathic effect index, it was concluded that under the same concentration treatment, the varieties with better seed germination had weaker autotoxicity inhibition effect. The germination potential, germination rate, germination index and vigor index of high-autotoxic variety “LZ” were significantly lower than those of low-autotoxic variety “656”, and the autotoxicity inhibition of germination potential and germination index was higher than that of 656. Similar to seed germination, the effect of extract on seedling growth differed between varieties, and the inhibitory effect of LZ on root length was greater than that of 656. It indicated that high-autotoxic alfalfa was more sensitive to autotoxicity stress, resulting in lower seed germination potential and germination rate than low autotoxic varieties. Consistent with previous studies, alfalfa varieties are one of the factors affecting the difference in autotoxicity [33,34,35]. Rong et al. treated seeds and seedlings of five varieties of alfalfa with extracts and found that there were significant differences in germination potential, germination rate, germination index, vigor index, radicle length, germ length and root activity of different alfalfa varieties under the same concentration treatment [6].

In addition, the occurrence of autotoxicity depends on the concentration of autotoxic substances, and the autotoxicity sensitivity thresholds of different alfalfa varieties are different [23]. In this experiment, the autotoxicity of high-autotoxic alfalfa LZ and low-autotoxic alfalfa 656 on germination rate changed from promotion to inhibition at C_0.01_ and C_0.025_, respectively. The inhibition concentration of LZ on germination rate was lower than that of 656. After the extract treatment, the different trends of high- and low-autotoxic alfalfa also indicated that their inhibition concentration thresholds were different. With increasing concentration, the seed germination potential and germination index of 656 showed a trend of increasing first and then decreasing, which showed that “low concentration promoted high concentration inhibition”, and the promotion effect of the extract gradually decreased, and the inhibition effect gradually increased. However, the seed germination potential and germination index of LZ gradually decreased, and the autotoxicity inhibition effect gradually increased. Therefore, when the concentration is low, low-autotoxic alfalfa was promoted, and high-autotoxic alfalfa was inhibited, indicating that the seed germination and seedling growth of high-autotoxic alfalfa are more susceptible to autotoxicity stress. Consistent with the research results of Zhang et al., alfalfa can be divided into susceptible and tolerant varieties, the response of seed germination to the intensity of autotoxicity was different in different varieties [36].

#### 3.1.2. The Physiological Response of Alfalfa Seedlings to Extracts Treatment

Osmotic adjustment ability

Autotoxicity is actually an environmental stress. It not only affects seed germination and seedling growth and development of offspring, but also activates the physiological response of seedlings to autotoxicity, mainly including osmotic regulation, initiation of the protective enzyme system, ROS metabolism, and alteration of membrane structure and function [37,38]. Under certain environmental stress, plant cells actively accumulate solutes, reduce osmotic potential and water potential, maintain turgor pressure, change the content of osmotic adjustment substances, and resist external stress through osmotic adjustment [39,40]. SS, SP and Pro are important osmotic regulators that can regulate and stabilize the cell osmotic potential, protect the cell structure to resist the damage caused by autotoxicity stress [41,42], and are closely associated with the ability of plants to resist autotoxicity [43,44]. This study found that under the stress of autotoxic substances, the content of osmotic adjustment substances such as SS, SP and Pro in the branchlets of *Casuarina equisetifolia* seedlings tend to increase and then decrease, and the increase was positively correlated with the stress concentration [45]. Li found that the content of Pro increased first and then decreased with increasing extracts concentration from metasequoia leaves [46]. The results of this experiment showed that with the increase in the extract concentration, the SS and Pro contents of the low-autotoxic variety 656 and the high-autotoxic variety LZ all showed a trend of increasing-decreasing-increasing, but in general, they were significantly higher than the control, and the promotion effect decreased first and then increased. It indicated that the increase in SS and Pro content in alfalfa was induced by autotoxicity stress, which may be a compensatory response to autotoxicity stress to maintain the osmotic potential level of cells and cell structure, thus improving the alfalfa ability to resist autotoxicity. With increasing concentration of the extract, the degree of autotoxicity stress deepened, resulting in an imbalance in its defense, which reduced the content of SS and Pro. However, in order to maintain normal growth and development, sufficient SS and Pro are needed to promote the accumulation of SS and Pro. Consistent with the results of previous studies, Zhang et al. found that with increasing concentration of autotoxic substabces, the SS content of alfalfa showed an increasing-decreasing-increasing trend, and the pro content was positively correlated with resistance [23]. In addition, soluble proteins is also an important osmotic adjustment substance in plants under stress [47]. In this experiment, the SP content of the low-autotoxic variety 656 showed a trend of decreasing first and then increasing, and the inhibitory effect gradually decreased. At C_0.01_, the inhibitory effect was turned into a promoting effect. It shows that autotoxicity stress can inhibit protein synthesis and even alleviate the inhibitory effect of stress on plant metabolism by degrading some of the synthesized proteins. The reason may be that the relative rate of proteins synthesis decreases and the existing soluble proteins are broken down in large quantities into free amino acids, which are used to regulate osmotic potential and provide metabolic energy, resulting in a decrease in soluble protein content. With the increase in stress degree, synthesized proteins are insufficient to maintain the inhibitory effect caused by stress, and osmotic potential is maintained by promoting the synthesis and accumulation of soluble proteins. Consistent with the results of previous studies, Li et al. studied the physiological response of Ginkgo biloba seedlings to the addition of Metasequoia leaf extract, and found that the SP content in Ginkgo biloba leaves was significantly lower than that of the control under low concentration extract treatment, but with the increase in the addition concentration, the SP content gradually increased [46]. The SP content of the high-autotoxic variety LZ increased first and then decreased, and the promotion effect changed to inhibition at C_0.005_. Consistent with the results of Zhang, the soluble protein content of autotoxic sensitive alfalfa showed a trend of increasing first and then decreasing, and the autotoxic-tolerant alfalfa showed a trend of decreasing first and then increasing [36].

Among the varieties, the SS content of low-autotoxic alfalfa 656 was higher than that of high-autotoxic alfalfa LZ, the SP content of 656 was higher than that of LZ under the treatment of high concentration extract, and the change trend of soluble protein of LZ and 656 was different after autotoxicity stress. The soluble protein content of LZ increased first and then decreased, 656 decreased first and then increased. It indicated that the accumulation of soluble sugar and soluble protein in high and low autotoxic varieties was different under autotoxicity stress, which was consistent with the results of previous studies. Chen et al. found that under the stress of cinnamic acid and vanillin, the osmotic adjustment ability of grafted eggplant was stronger than that of own-rooted eggplant, and the contents of soluble sugar, soluble protein and proline in grafted eggplant were higher than those in own-rooted eggplant [48]. Yang et al. found that the stress of o-hydroxybenzoic acid showed a promoting effect on the content of free amino acids in Chinese fir clones. No.01 has a strong ability to resist autotoxic substances stress and has a strong promoting effect [49]. In addition, this experiment also found that the osmotic adjustment substance content of varieties with strong stress resistance can be maintained at a high level under autotoxic stress, reducing the degree of cell membrane damage. Consistent with the results of Zhang et al., the changes in osmotic adjustment substances in cells are related to the strength of plant resistance and sensitivity. Autotoxic-tolerant alfalfa has stronger osmotic adjustment ability, and the contents of soluble sugar, soluble protein and free proline are higher than those of autotoxic sensitive alfalfa [36].

In general, the change trend of different osmotic adjustment substances may have different responses to the stress intensity of autotoxic substances, but in the process of osmotic adjustment, proline, soluble sugar and soluble protein complement each other and coordinate to better alleviate the osmotic pressure caused by stress [50].

Antioxidant system

During normal plant growth and development, the production and scavenging of free radicals in the body is in a dynamic equilibrium. Stress reduces the activity of antioxidant enzymes such as POD, SOD, and CAT and accumulates a large amount of reactive oxygen species such as O^2−^, H_2_O_2_ and OH^−^, resulting in the peroxidation of unsaturated fatty acids in the cell membrane and their gradual decomposition into MDA, aggravating the degree of membrane lipid peroxidation and changing the permeability of cell membrane [51]. Autotoxicity mainly damages plants mainly by destroying the redox balance [52]. The activities of antioxidant defense enzymes such as SOD, POD and CAT in plants were induced by autotoxicity stress. SOD is the first to play a role in ROS scavenging, specifically scavenging O^2−^, thereby ensuring a lower level of ROS [53]. POD and CAT are mainly responsible for scavenging H_2_O_2_ and reducing the accumulation of H_2_O_2_, thereby reducing the damage of ROS to plants [54]. Li et al. found that the activities of SOD, POD and CAT in grass leaves were not significantly different from those in the control or increased under the treatment of low amount of *Eucalyptus gradis* leaf litter and short time, but with increasing treatment time and amount, the activities of the three enzymes decreased [25]. The results of this experiment showed that the changes in antioxidant enzyme activity were different according to the degree of stress and the alfalfa variety. The comparison between the different treatment concentrations showed that the POD activity of seedlings of the high-autotoxic variety 656 and the low-autotoxic variety LZ was significantly higher than that of the control. With increasing concentration, POD activity first increased and then decreased, SOD activity showed an increasing-decreasing-increasing trend, and the SOD activity was lowest at C_0.005_. The CAT activity of 656 gradually increased, LZ showed an increase–decrease–increase trend, but both were significantly higher than that of the control. It is indicated that 656 and LZ can remove H_2_O_2_ in time and maintain the relative balance of ROS by increasing the activity of antioxidant enzymes (POD, SOD, CAT) under low concentration treatment, so as to reduce the damage caused by ROS accumulation and ensure their normal metabolism. As the stress level increased, the activity of POD and SOD decreased, and the ability of alfalfa to remove excess ROS decreased, leading to an accumulation of ROS content in plants, resulting in oxidative stress. In agreement with the results of previous studies. Zhao et al. found that low concentrations of Astragalus mongholicus root exudates had no inhibitory effect on seed germination and seedling growth, and high concentrations of root exudates inhibited the activity of SOD, POD and CAT in Astragalus mongholicus seedlings, thus affecting the growth of seedlings [55]. In this experiment, in order to reduce the oxidative stress of the plants, the activity of SOD and CAT increased as the level of stress increased, when the concentration was higher than C_0.005_. In addition, the POD activity of 656 was higher than that of LZ, and the increase in 656 was higher than that of LZ compared with the control. The results showed that the antioxidant capacity of low-autotoxic variety 656 was higher than that of high-autotoxic variety LZ, and it had stronger oxidative scavenging ability. The POD activity of 656 and LZ began to decrease at C_0.005_ and C_0.001_, respectively, indicating that the inhibition concentration of POD activity of high-autotoxic alfalfa and low-autotoxic alfalfa was different. The inhibition concentration of 656 was lower than that of LZ, and it was more sensitive.

The accumulation of reactive oxygen induced by oxidative damage has been proved to be an important factor causing autotoxicity in plants. Excessive production of reactive oxygen leads to a variety of harmful cellular effects including lipid peroxidation, protein denaturation, nucleic acid degradation and photosynthetic obstruction of biofilms, thereby increasing cell membrane permeability and destroying cell function [21]. The accumulation of H_2_O_2_ in plants has a dual function, not only as a signal substance to induce plant defense responses, but also its excessive accumulation can lead to oxidative damage in plants [56]. MDA content reflects the degree of cell membrane lipid peroxidation [57]. Studies have shown that membrane destruction by autotoxic substances may be the starting point of the autotoxicity, and that disruption of the normal structure and function of the membrane is a mode of action of autotoxic substances [58]. Autotoxicity stress severely inhibits the root growth of melon seedlings, induces the accumulation of reactive oxygen species in roots and depolarization of root cell membrane [21]. Hu et al. found that rice straw extract could inhibit the metabolic function of reactive oxygen species in seedlings [59]. The water extract from stem and leaves of Artemisia frigida significantly increased the MDA content in seedlings [60]. The results of this experiment showed that the MDA content and H_2_O_2_ content of low-autotoxic variety 656 and high-autotoxic variety LZ were higher than those of the control, and increased first and then decreased with increasing treatment concentration. It was shown that after treatment with the extract, MDA and H_2_O_2_ accumulated in alfalfa seedlings, the cell membrane system was damaged and the degree of membrane lipid peroxidation was aggravated. There were also differences in sensitive concentrations between the low and high autotoxic varieties. The contents of MDA and H_2_O_2_ in 656 and LZ were at the highest levels at C_0.001_ and C_0.005_, respectively, indicating that 656 was more sensitive to autotoxicity stress than LZ. The production rate of ROS was lower, and the accumulation of reactive oxygen species could stimulate the antioxidant protection mechanism earlier. This is also the reason why the active oxygen content of 656 is lower than that of LZ. Zhang et al. have shown that autotoxicity can induce oxidative stress in mitochondria, which is manifested as the accumulation of ROS. The contents of MDA and H_2_O_2_ in autotoxic sensitive alfalfa are higher than those in autotoxic-tolerant alfalfa, and the sensitive concentrations of the two to the rapid response of autotoxicity are different [23], which is consistent with the results of this study. High-autotoxic alfalfa produces more MDA and H_2_O_2_, which destroys the oxidation balance and leads to oxidative damage, while low-autotoxic alfalfa can stimulate its oxidation scavenging system to remove excess ROS, thus maintaining its redox dynamic balance.

### 3.2. The Relationship between the Key Autotoxic Substances and Autotoxicity of Alfalfa

Phenolic secondary metabolites are the most abundant class of autotoxic substances isolated and identified from plants [8,61]. In recent years, studies have been carried out on the variation in the levels of single autotoxic substances in plants and the autotoxicity they produce. Yeasmin et al. investigated the effects of potential allelochemicals on the growth and nutrient absorption of two asparagus varieties. They found that UC157 produced higher concentrations of total allelochemicals than Gijnlim, and the two varieties showed significant differences in growth, nutrient absorption and allelopathic effects [62]. Different genotypes of cassava root exudates have different chemical substances and relative content, resulting in different resistance to continuous cropping obstacles in different varieties [63]. Studies have shown that there are autotoxic-tolerant and sensitive varieties of alfalfa, and that varieties with a high content of primary autotoxic substances in the extract may be more sensitive to autotoxic stress [23]. In this study, the content of phenolic autotoxic substances in alfalfa was determined, and the results showed that there were significant differences in the content of autotoxic substances in different alfalfa varieties. The content of autotoxic substances was highest in LZ and lowest in 656. The comparative analysis of the differences in the content of single autotoxic substances in the same variety of alfalfa showed that the total coumarin content was the highest in eight varieties of alfalfa, followed by ferulic acid and cinnamic acid, and the content of p-hydroxybenzoic acid and vanillic acid was low, which was consistent with the results of previous studies. Rong and Li et al. measured the content of main phenolic acids in the aboveground and underground parts of different alfalfa varieties, and found that the content of coumarin was the highest [6,22]. In addition, by sorting the comprehensive scores of the autotoxicity of eight alfalfa varieties, the alfalfa variety with the strongest autotoxicity was “LZ”, and the alfalfa variety with the weakest autotoxicity was “656”. In general, LZ had the highest levels of autotoxic substances and the strongest autotoxicity. 656 had the lowest level of autotoxic substances and the weakest autotoxicity. Coumarin is an autotoxic substance that plays a major role in alfalfa. This is consistent with previous research. Zheng et al. found that the inhibitory effects of exogenous autotoxic substances coumarin, cinnamic acid and hydroxybenzoic acid on the growth and development of alfalfa were different, with coumarin being the strongest [64]. Song et al. found that the autotoxicity of coumarin was stronger than that of coumaric acid through the experiments on alfalfa seed germination and seedling growth [65]. Tao et al. found that the autotoxicity inhibition effect of coumarin on alfalfa seed germination was significantly higher than that of other phenolic acids by exogenous addition of five autotoxic substances. The inhibitory effect increased with increasing treatment concentration.

In order to further clarify the relationship between coumarin, autotoxic substances and autotoxicity in alfalfa, the correlation analysis of single autotoxic substances, total autotoxic substances and autotoxicity was carried out in this study. It was found that the autotoxicity was significantly positively correlated with the content of total autotoxic substances and total coumarin, and the content of total coumarin was significantly positively correlated with the content of total autotoxic substance. Subsequently, the contribution of a single autotoxic substance to the autotoxicity was analyzed. The eight varieties of alfalfa showed that the total coumarin had the greatest contribution to the content of autotoxic substances, and the contribution rate was higher than 96.00%, which further indicated that coumarin was the main component of alfalfa autotoxic substances. The higher the coumarin content, the higher the total autotoxic substance content, and the stronger the autotoxicity.

## 4. Materials and Methods

### 4.1. Plant Material

The *Medicago sativa*, such as Gan-nong No.9 (G9), WL656HQ (656), Xin-mu No.4 (X4), Longzhong (LZ), MF4114 (4114), MF4010 (4010), Qingshui (QS) and 3105c (3105) were used for the experiment. The seeds of G9, LZ, 4114, 4010 and 3105 were provided by the Key Laboratory of Grassland Ecosysem of Ministry of Education in Gansu Agricultural Univerity, the seed of 656 was provided by Beijng Rytway, the seed of X4 and QS were provided by Xinjiang Agricultural University.

### 4.2. Growth Conditions and Treatments

The experiment was carried out from April to June 2022 at the experimental base of Gansu Agricultural University (GAU; 34°05′ N, 105°41′ E, 1525 m altitude), Lanzhou, northwestern China. In the completely randomized block design, eight alfalfa varieties were treated with eight treatments, and each treatment was randomly placed in three pots and repeated three times. The nutrient soil, turfy soil and vermiculite manure (2:1:1) were mixed evenly and put into the plastic pots with a size of 25 cm × 20 cm (outer diameter × deep). Then, 2/3 of the flowerpots were embedded in soil under field environment. The full and uniformly sized alfalfa seeds were selected, sterilized with 10% sodium hypochlorite solution and plant into plastic pots. After seedling emergence, the consistent and uniform growth of 10 seedlings each plot was cultivated by watering 300 mL per 2–3 days and 100 mL Hoagland’s solution per week which is required to guarantee normal growth. The leaves of alfalfa were obtained at the initial flowering stage and brought back to the laboratory. After drying and crushing in the shade, they were passed through a 1 mm sieve and stored at room temperature for the preparation of the extracts and the determination of phenolic content.

### 4.3. Preparation of Extracts from the Leaves of Alfalfa

An amount of 5 g of the sample was accurately weighed and placed in a triangular flask with 100 mL of distilled water, sealed with a sealing membrane, extracted at 210 r·min^−1^ in a shaker at 25 °C for 48 h, filtered by double-layer gauze, centrifuged at 3000 r·min^−1^ at room temperature for 15 min, and the supernatant was 0.05 g·mL^−1^ extract. Extract with concentraction of 0.025 g·mL^−1^, 0.01 g·mL^−1^, 0.005 g·mL^−1^ and 0.001 g·mL^−1^ (denoted as C_0.05_, C_0.025_, C_0.01_, C_0.005_ and C_0.001_, respectively) were obtained by dilution and placed in a refrigerator at 4 °C for use.

### 4.4. The Seed Germination Test

In the Petri dish filter paper method, the seeds of eight alfalfa varieties were treated with different concentrations of leaf extracts, and the sterile water treatment were used as the control. The full and uniform alfalfa seeds were selected and sterilized with 10% sodium hypochlorite for 10 min, Then, rinse three times with distilled water and six times with sterile water., and dried with filter paper. Two layers of filter paper were spread on a culture dish with a diameter of 9 cm, and 30 seeds were placed neatly in each culture dish. Different concentration extracts of 5 mL were added (the extract was placed at room temperature for 3–5 h before use to avoid the effect of low temperature on seed germination), and 5 mL sterile water was added to the control (CK). Each treatment was repeated four times. The culture dish was covered and placed in an artificial incubator. The temperature was set at 25 °C, the illumination time was 12 h, and the illumination intensity was 2000 lx. From the first day of germination, the number of germinations was counted per 24 h, and 2 mL extract or sterile water was supplemented per 48 h.

### 4.5. Measurement Indexes

#### 4.5.1. Determination of Phenolic Autotoxic Substances in Alfalfa Leaf Extracts

Determination of chlorogenic acid, cinnamic acid, p-coumaric acid, caffeic acid, ferulic acid, p-hydroxybenzoic acid and vanillic acid content

Preparation of the standard solution: 5 mg of chlorogenic acid, cinnamic acid, p-coumaric acid, caffeic acid, ferulic acid, p-hydroxybenzoic acid and vanillic acid were accurately weighed (all purchased from Shanghai Yuanye Biotechnology Co., Ltd., Shanghai, China), and the volume was adjusted to 10 mL with chromatographic methanol, and then 1 mL of the above standard solution was mixed to obtain the mixed standard solution stock solution. The standard was identified by analyzing the peak time. The regression equation of each standard was obtained by taking the peak area as the ordinate (Y) and the standard concentration as the abscissa (X, μg·mL^−1^).

Extraction of phenolic compounds from alfalfa: 0.5 g sample was added with 10 mL of pre-cooled 80% methanol and refrigerated at 4 °C for 1 h. After ultrasonic extraction for 30 min, it was centrifuged at 10,000 rpm for 10 min at 4 °C. Finally, the supernatant was obtained by filtration with 0.22 μm filter membrane [66].

Determination of phenolic content in alfalfa: The contents of chlorogenic acid, caffeic acid, p-hydroxybenzoic acid, vanillic acid, ferulic acid, p-coumaric acid and trans-cinnamic acid in alfalfa leaves were determined by High performance liquid chromatography (HPLC). The chromatographic conditions were set as follows: Eclipse Plus C_18_ column (4.6 mm × 250 mm, 5 μm, Waters Symmetry), flow rate: 1.0 mL·min^−1^, column temperature: 30 °C, injection volume: 10 μL, detection wavelength: 275 nm. Mobile phase: 0.5% glacial acetic acid (A) and acetonitrile (B) gradient elution, the elution procedure was (0~15 min, 15~25% B; 15~18 min, 25~40% B; 18~22 min, 40~15% B; 22~30 min, 15% B). A total of 3 repetitions were carried out.

Determination of total coumarin content

Determination by spectrophotometry. The extraction of total coumarin was slightly improved according to the method of Tang Chunni [67]. A total of 10 mL 60% ethanol was added to the sample (0.5 g). After shaking well, the sample was ultrasonically extracted at 30 °C for 40 min. After centrifugation at 10,000 rpm for 5 min at 4 °C, the supernatant was taken and repeatedly extracted for 3 times. The filtrate was combined and the absorbance was measured at 374 nm. A total of 3 repetitions were carried out.

#### 4.5.2. Determination of Seed Germination and Seedling Growth Indexes

The number of germinated seeds was recorded regularly every day, and the germination standard was 2 mm of radicle breaking through the seed coat. Germination potential (GP) was calculated on the 3rd day of seed germination. On the 7th day of seed germination, the germination rate (GR) and germination index (GI) of seeds were calculated, the root length (RL), seedling length (SL), fresh weight (FW) and dry weight (DW) of seedlings were measured, the vigor index (VI) and response index (RI) of each index were calculated. The above indicators were repeated 4 times. The calculation formula is as follows:GP = Number of germinated seeds on the third day/Total number of seeds tested × 100%,(2)
GR = Number of germinated seeds on the 7th day/Total number of seeds tested × 100%,(3)
GI = Σ Number of different days germinated/The corresponding germination days,(4)
VI = Germination index × (Root length + Seedling length),(5)
RI = 1 − C/T (T ≥ C) or RI = C/T − 1(T < C),(6)

In the formula: T is the treatment value; C is the control value; when RI ≥ 0, there is a promotion effect; when RI < 0, there is a suppression effect.

#### 4.5.3. Determination of Seedling Physiological Indexes

Antioxidant index: Peroxidase (POD) activity was determined by guaiacol method [68]. Superoxide dismutase (SOD) activity was determined by nitrogen blue tetrazolium photoreduction method [69]. Catalase (CAT) activity was measured using a kit (purchased from Suzhou Grace Biological Co., Ltd. (Suzhou, China)).

Osmotic adjustment substances: Free proline (Pro) content was determined by acid ninhydrin method [69]. Soluble sugar (SS) content was determined by anthrone colorimetry [68]. Soluble protein (SP) content was determined by Coomassie brilliant blue G-250 staining [70].

Membrane lipid peroxide: Malondialdehyde (MDA) content was determined by thiobarbituric acid colorimetric method [71]. Hydrogen peroxide (H_2_O_2_) content was determined by kit (purchased from Suzhou Grace Biological Co., Ltd.).

The determination of POD was repeated 4 times, other physiological indexes was repeated 3 times.

### 4.6. Statistical Analysis

Microsoft Excel 2016 was used for data collation and calculation, and Origin 2023 was used for drawing. Statistical analyses were assessed by one-way ANOVA followed by Duncan’s multiple range test (*p* < 0.05) using SPSS 20.0 and comparisons between the mean values were performed using least square difference (LSD) test at 5% probability level.

## 5. Conclusions

The content of phenolic autotoxic substances in eight varieties of alfalfa was significantly different, among which the content of total coumarin was the highest, whereas that of ferulic acid, vanillic acid and p-hydroxybenzoic acid was lower. The variety with the highest total coumarin content was LZ, and the lowest was 656.

This experiment preliminarily clarified the phenotypic and physiological differences in different varieties of alfalfa after treatment with the extract, and obtained alfalfa varieties with significant differences in terms of autotoxic effects. (1) Compared with LZ, 656 had stronger osmotic regulation and antioxidant capacity, lower membrane lipid peroxidation and ROS accumulation after treatment with extracts. (2) With increasing extract concentration, the germination and physiological indexes showed “low promotion and high inhibition”. (3) The principal component analysis of the autotoxicity of eight alfalfa varieties showed that the alfalfa variety with the strongest autotoxicity was “LZ”, and the weakest autotoxicity was “656”. The comprehensive scores of LZ after extract treatment were lower than those of the control, and the comprehensive scores of 656 were higher than those of the control.

By analyzing the relationship between the content of autotoxic substances and autotoxicity, and the contribution of single autotoxic substances to autotoxicity, it was found that total coumarin has the greatest contribution to the content of autotoxic substances and is the main component of alfalfa autotoxic substances. Therefore, the higher the content of total coumarin, the higher the content of autotoxic substances, and the stronger the autotoxicity.

The high-autotoxic variety “LZ” and the low-autotoxic variety “656” with different total coumarin content and autotoxicity effect were screened. Therefore, from the perspective of autotoxicity, 656 is more suitable for alfalfa ecological grassland construction, but further field experiments are needed to verify this.

## Figures and Tables

**Figure 1 plants-12-03263-f001:**
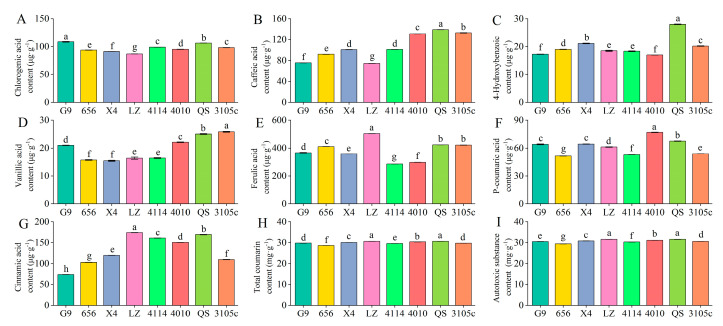
Differences in phenolic autotoxic substance content in extracts of different alfalfa varieties. (**A**) Chlorogenic acid content; (**B**) caffeic acid content; (**C**) 4-hydroxybenzoic content; (**D**) vanillic acid content; (**E**) ferulic acid content; (**F**) P-coumaric acid content; (**G**) cinnamic acid content; (**H**) total coumarin content; (**I**) autotoxic substance content. Different lowercase letters indicate significant differences between different varieties at the *p* < 0.05 level.

**Figure 2 plants-12-03263-f002:**
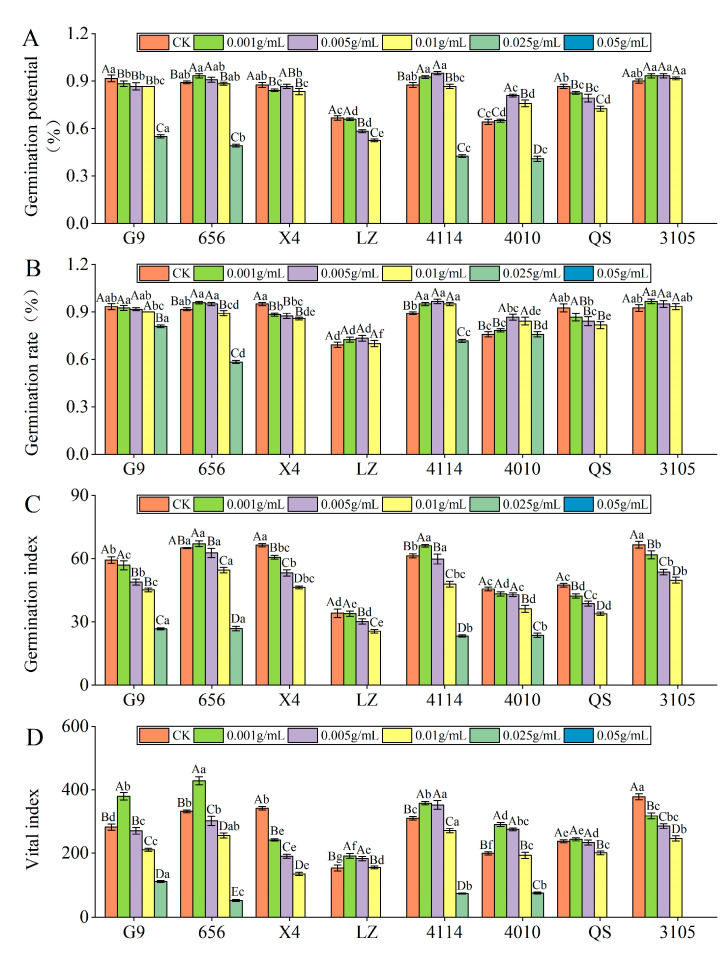
Effect of alfalfa extracts on its seed germination. (**A**) Germination potential; (**B**) germination rate; (**C**) germination index; (**D**) vital index. Different capital letters indicate significant differences between different treatment concentrations of the same variety (*p* < 0.05). Different lowercase letters indicate significant differences between varieties of the same treatment concentration (*p* < 0.05).

**Figure 3 plants-12-03263-f003:**
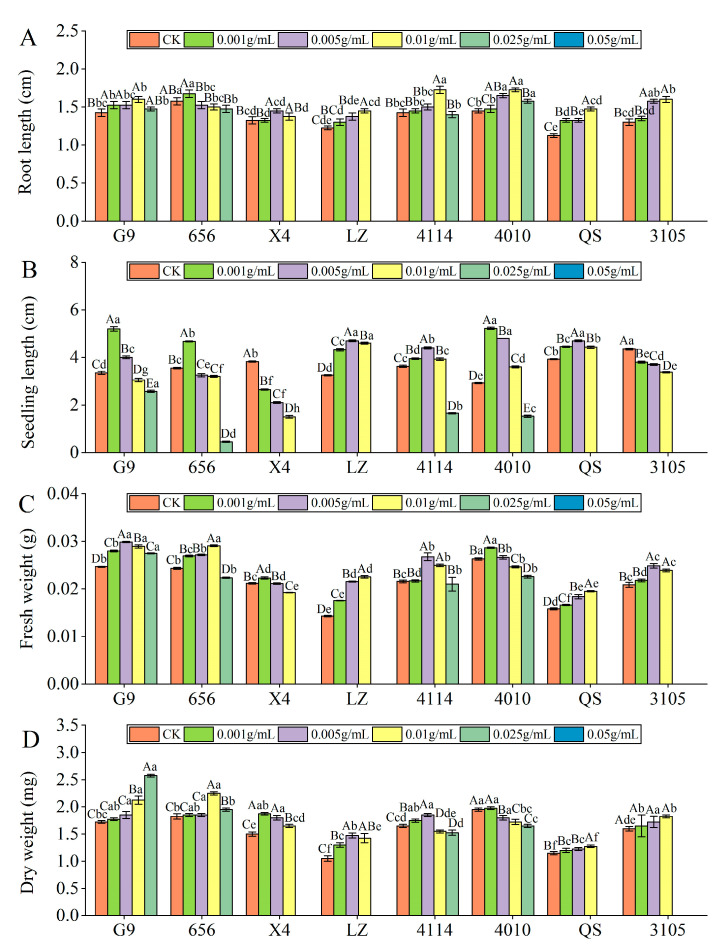
Effect of alfalfa extracts on its seedling growth. (**A**) Root length; (**B**) seedling length; (**C**) fresh weight; (**D**) dry weight. Different capital letters indicate significant differences between different treatment concentrations of the same variety (*p* < 0.05). Different lowercase letters indicate significant differences between varieties of the same treatment concentration (*p* < 0.05).

**Figure 4 plants-12-03263-f004:**
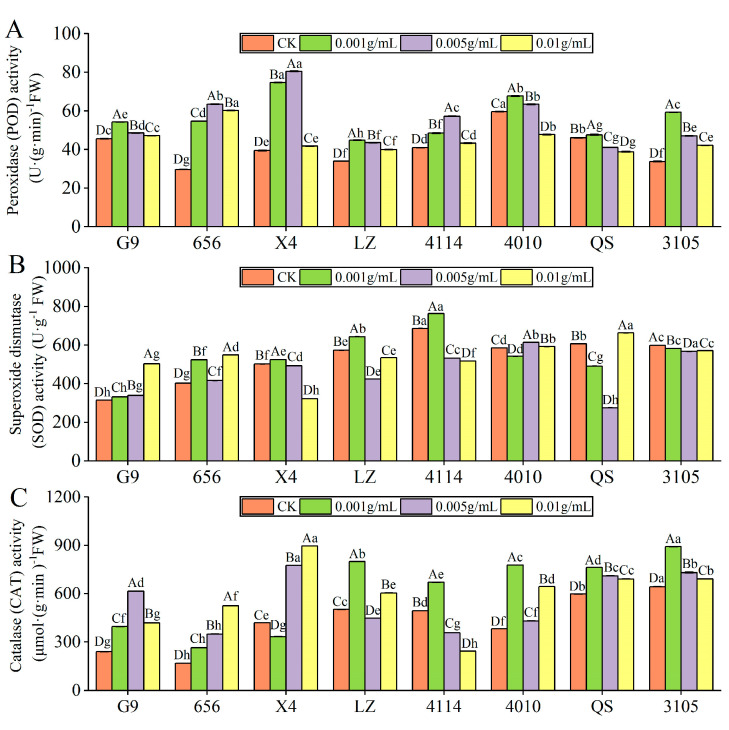
Effect of alfalfa extracts on the activity of antioxidative enzyme in seedlings. (**A**) Peroxidase activity; (**B**) superoxide dismutase activity; (**C**) catalase activity. Different capital letters indicate significant differences between different treatment concentrations of the same variety (*p* < 0.05). Different lowercase letters indicate significant differences between varieties of the same treatment concentration (*p* < 0.05).

**Figure 5 plants-12-03263-f005:**
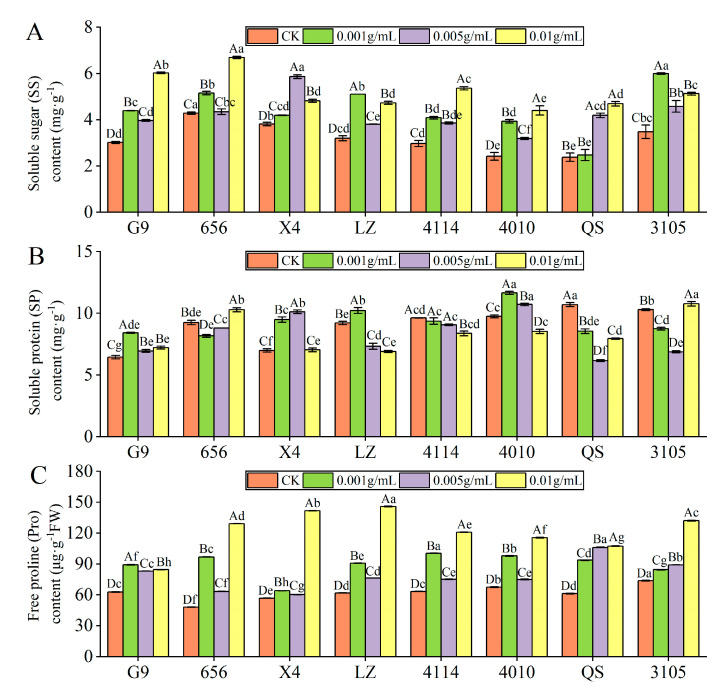
Effect of alfalfa extracts on osmotic regulation substance content in seedlings. (**A**) Soluble sugar content; (**B**) soluble protein content; (**C**) free proline content. Different capital letters indicate significant differences between different treatment concentrations of the same variety (*p* < 0.05). Different lowercase letters indicate significant differences between varieties of the same treatment concentration (*p* < 0.05).

**Figure 6 plants-12-03263-f006:**
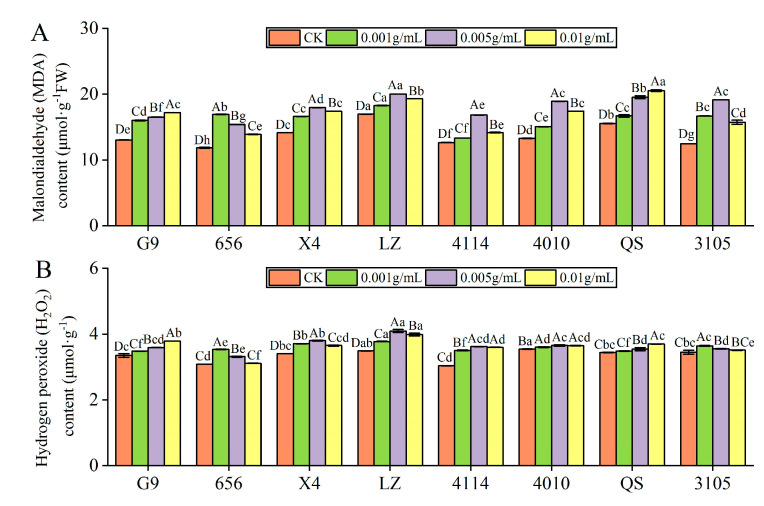
Effect of alfalfa extracts on malondialdehyde and hydrogen peroxide content in seedlings. (**A**) Malondialdehyde content; (**B**) hydrogen peroxide content. Different capital letters indicate significant differences between different treatment concentrations of the same variety (*p* < 0.05). Different lowercase letters indicate significant differences between varieties of the same treatment concentration (*p* < 0.05).

**Figure 7 plants-12-03263-f007:**
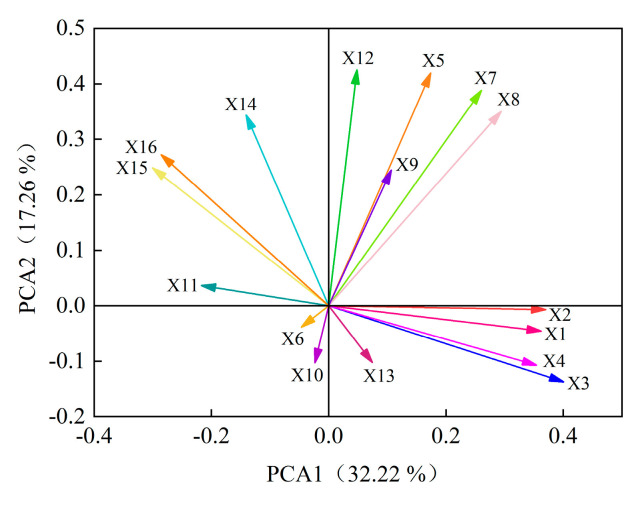
Rotated component of principal component analysis. Germination potential (X1), germination rate (X2), germination index (X3), vital index (X4), root length (X5), seedling length (X6), fresh weight (X7), dry weight (X8), peroxidase activity (X9), superoxide dismutase activity (X10), catalase activity (X11), soluble sugar content (X12), soluble protein content (X13), free proline content (X14), malondialdehyde content (X15), and hydrogen peroxide content (X16).

**Figure 8 plants-12-03263-f008:**
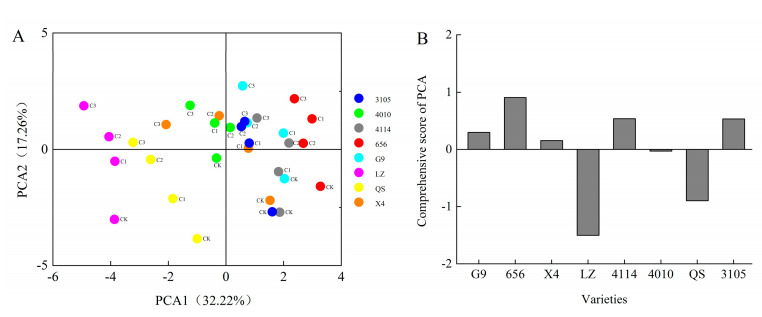
The comprehensive scores and ranking of different varieties of alfalfa under the extract treatment. (**A**) The comprehensive score of alfalfa after extract treatment. Each color represents a variety of alfalfa. CK represents the extract concentration of 0 g·mL^−1^, C1 represents the extract concentration of 0.001 g·mL^−1^, C2 represents the extract concentration of 0.005 g·mL^−1^, and C3 represents the extract concentration of 0.01 g·mL^−1^. (**B**) Autotoxicity ranking of different varieties of alfalfa.

**Figure 9 plants-12-03263-f009:**
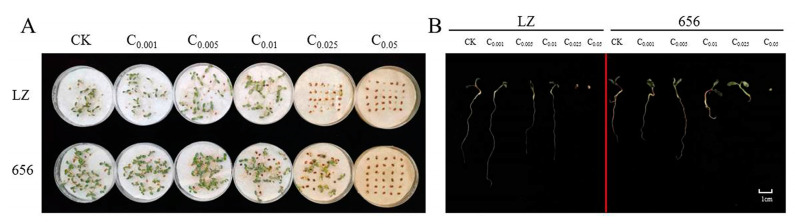
(**A**) Germination of high-autotoxic alfalfa “LZ” and low-autotoxic alfalfa “656” after treatment with extract. (**B**) The seedling growth of high-autotoxic alfalfa “LZ” and low-autotoxic alfalfa “656” after extract treatment.

**Figure 10 plants-12-03263-f010:**
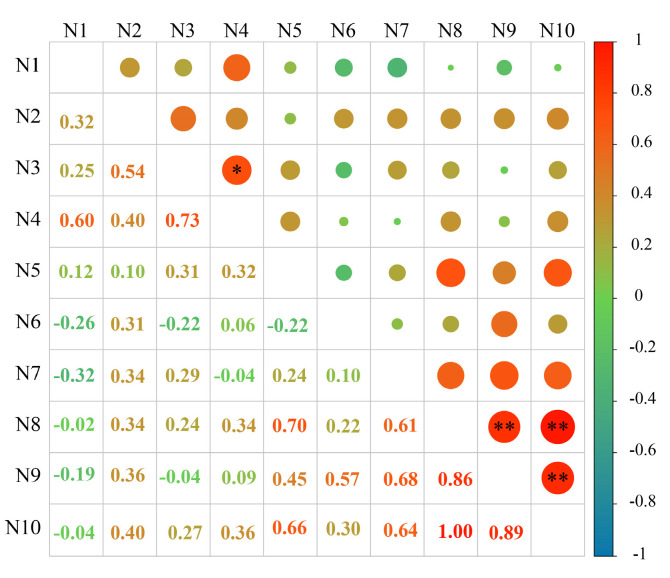
Correlation analysis between the content of phenolic autotoxic substances and autotoxicity in alfalfa. Chlorogenic acid content (N1), p-hydroxybenzoic acid content (N2), caffeic acid content (N3), vanillic acid content (N4), p-coumaric acid content (N5), ferulic acid content (N6), trans-cinnamic acid content (N7), coumarins content (N8), comprehensive scores (N9), and autotoxic substance content (N10). * represents a significant correlation at the *p* < 0.05 level, and ** represents a extremely significant correlation at the *p* < 0.01 level.

**Figure 11 plants-12-03263-f011:**
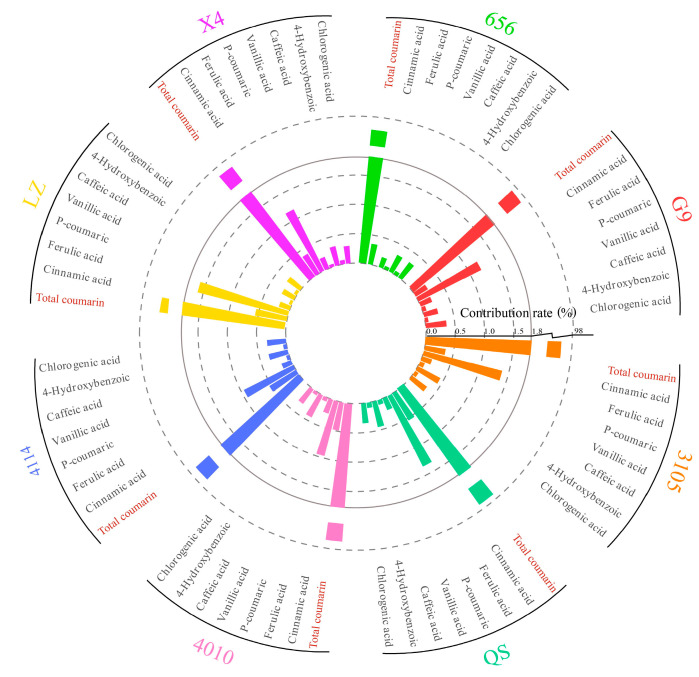
Contribution rate of single autotoxic substances in alfalfa to autotoxicity.

**Figure 12 plants-12-03263-f012:**
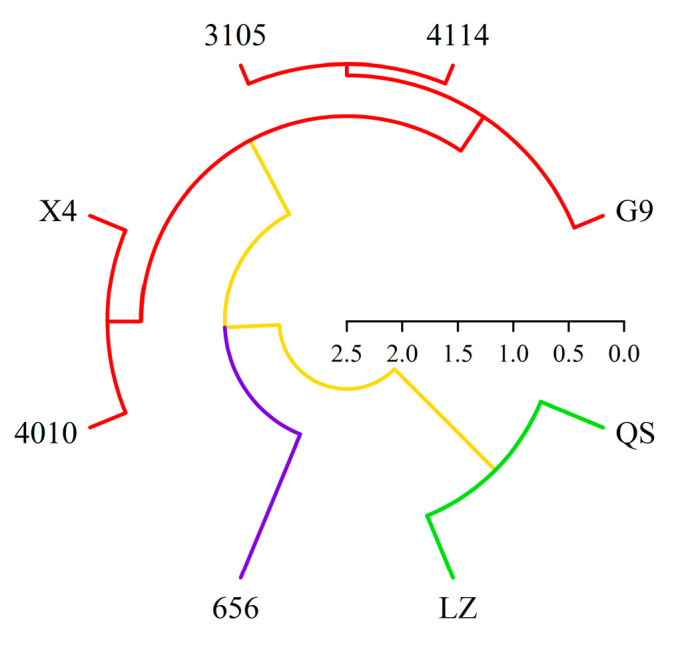
Cluster diagram of different varieties of alfalfa based on total coumarin content and autotoxicity. Red represents medium autotoxic variety, blue represents low-autotoxic variety, green represents high-autotoxic variety.

**Figure 13 plants-12-03263-f013:**
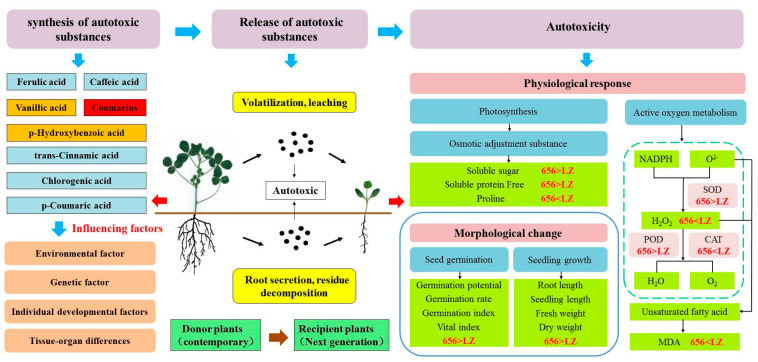
Autotoxicity process model of alfalfa.

**Table 1 plants-12-03263-t001:** Allelopathic effect index of alfalfa extracts on seed germination and seedling growth index.

Forage Extracts		Response Indexes (RI)							
Varieties	Concentrations	Seed Germination Indexes				Seedling Growth Indexes			
		GP	GR	GI	VI	RL	SL	FW	DW
G9	C_0.001_	−0.04	−0.01	−0.04	0.26	0.35	0.07	0.12	0.03
	C_0.005_	−0.05	−0.02	−0.18	−0.04	0.16	0.08	0.17	0.08
	C_0.01_	−0.05	−0.04	−0.24	−0.25	−0.08	0.12	0.15	0.19
	C_0.025_	−0.40	−0.13	−0.55	−0.61	−0.22	0.06	0.10	0.34
	C_0.05_	−1.00	−1.00	−1.00	−1.00	−1.00	−1.00	−1.00	−1.00
656	C_0.001_	0.04	0.04	0.03	0.23	0.25	0.08	0.10	0.01
	C_0.005_	0.02	0.04	−0.04	−0.09	−0.08	−0.02	0.10	0.02
	C_0.01_	−0.01	−0.03	−0.16	−0.23	−0.10	−0.04	0.16	0.19
	C_0.025_	−0.45	−0.36	−0.59	−0.85	−0.87	−0.07	−0.08	0.06
	C_0.05_	−1.00	−1.00	−1.00	−1.00	−1.00	−1.00	−1.00	−1.00
X4	C_0.001_	−0.04	−0.07	−0.09	−0.29	−0.31	0.02	0.05	0.20
	C_0.005_	−0.01	−0.08	−0.20	−0.44	−0.45	0.11	0.00	0.16
	C_0.01_	−0.05	−0.10	−0.30	−0.61	−0.61	0.06	−0.09	0.07
	C_0.025_	−1.00	−1.00	−1.00	−1.00	−1.00	−1.00	−1.00	−1.00
	C_0.05_	−1.00	−1.00	−1.00	−1.00	−1.00	−1.00	−1.00	−1.00
LZ	C_0.001_	−0.01	0.05	0.00	0.20	0.25	0.06	0.19	0.19
	C_0.005_	−0.13	0.06	−0.11	0.16	0.30	0.12	0.34	0.28
	C_0.01_	−0.21	0.01	−0.25	0.01	0.29	0.16	0.37	0.26
	C_0.025_	−1.00	−1.00	−1.00	−1.00	−1.00	−1.00	−1.00	−1.00
	C_0.05_	−1.00	−1.00	−1.00	−1.00	−1.00	−1.00	−1.00	−1.00
4114	C_0.001_	0.05	0.06	0.07	0.13	0.08	0.02	0.01	0.06
	C_0.005_	0.08	0.08	−0.02	0.12	0.17	0.05	0.19	0.11
	C_0.01_	−0.01	0.06	−0.22	−0.13	0.07	0.18	0.14	−0.07
	C_0.025_	−0.51	−0.20	−0.62	−0.77	−0.54	−0.01	−0.03	−0.08
	C_0.05_	−1.00	−1.00	−1.00	−1.00	−1.00	−1.00	−1.00	−1.00
4010	C_0.001_	0.01	0.03	−0.05	0.31	0.44	0.03	0.08	0.02
	C_0.005_	0.21	0.13	−0.06	0.28	0.39	0.11	0.01	−0.06
	C_0.01_	0.15	0.10	−0.21	−0.03	0.19	0.16	−0.06	−0.12
	C_0.025_	−0.36	0.00	−0.48	−0.63	−0.47	0.08	−0.14	−0.15
	C_0.05_	−1.00	−1.00	−1.00	−1.00	−1.00	−1.00	−1.00	−1.00
QS	C_0.001_	−0.05	−0.06	−0.11	0.03	0.12	0.16	0.05	0.03
	C_0.005_	−0.09	−0.09	−0.18	−0.02	0.17	0.16	0.14	0.05
	C_0.01_	−0.16	−0.12	−0.28	−0.15	0.12	0.25	0.19	0.09
	C_0.025_	−1.00	−1.00	−1.00	−1.00	−1.00	−1.00	−1.00	−1.00
	C_0.1_	−1.00	−1.00	−1.00	−1.00	−1.00	−1.00	−1.00	−1.00
3105	C_0.001_	0.04	0.04	−0.07	−0.16	−0.13	0.03	0.04	0.03
	C_0.005_	0.04	0.03	−0.19	−0.24	−0.14	0.17	0.16	0.06
	C_0.01_	0.02	0.01	−0.25	−0.35	−0.23	0.18	0.13	0.12
	C_0.025_	−1.00	−1.00	−1.00	−1.00	−1.00	−1.00	−1.00	−1.00
	C_0.05_	−1.00	−1.00	−1.00	−1.00	−1.00	−1.00	−1.00	−1.00

**Table 2 plants-12-03263-t002:** Allelopathic effect index of alfalfa extracts on physiological indexes of seedlings.

Forage Extracts		Response Indexes (RI)							
Varieties	Concentrations	POD	SOD	CAT	SS	SP	PRO	MDA	H_2_O_2_
G9	C_0.001_	0.16	0.05	0.39	0.31	0.24	0.30	0.18	0.04
	C_0.005_	0.06	0.07	0.61	0.24	0.07	0.25	0.21	0.07
	C_0.01_	0.03	0.37	0.43	0.50	0.11	0.26	0.24	0.11
656	C_0.001_	0.46	0.23	0.37	0.17	−0.12	0.50	0.30	0.13
	C_0.005_	0.53	0.03	0.52	0.01	−0.05	0.24	0.23	0.07
	C_0.01_	0.51	0.27	0.68	0.36	0.10	0.63	0.15	0.01
X4	C_0.001_	0.47	0.04	−0.21	0.09	0.26	0.11	0.15	0.08
	C_0.005_	0.51	−0.02	0.46	0.35	0.31	0.06	0.21	0.11
	C_0.01_	0.05	−0.36	0.53	0.21	0.01	0.60	0.19	0.07
LZ	C_0.001_	0.24	0.11	0.37	0.37	0.10	0.32	0.07	0.08
	C_0.005_	0.22	−0.26	−0.11	0.16	−0.20	0.19	0.15	0.15
	C_0.01_	0.15	−0.07	0.17	0.32	−0.25	0.58	0.12	0.12
4114	C_0.001_	0.15	0.10	0.27	0.27	−0.03	0.37	0.05	0.13
	C_0.005_	0.28	−0.22	−0.28	0.23	−0.06	0.16	0.25	0.16
	C_0.01_	0.05	−0.25	−0.51	0.45	−0.13	0.48	0.11	0.16
4010	C_0.001_	0.12	−0.08	0.51	0.38	0.16	0.31	0.12	0.02
	C_0.005_	0.06	0.05	0.11	0.24	0.09	0.10	0.30	0.03
	C_0.01_	−0.20	0.01	0.41	0.45	−0.13	0.42	0.24	0.03
QS	C_0.001_	0.03	−0.19	0.22	0.04	−0.20	0.35	0.07	0.01
	C_0.005_	−0.11	−0.55	0.16	0.43	−0.42	0.42	0.20	0.03
	C_0.01_	−0.16	0.08	0.13	0.49	−0.26	0.43	0.24	0.07
3105	C_0.001_	0.43	−0.03	0.28	0.42	−0.15	0.13	0.25	0.05
	C_0.005_	0.28	−0.05	0.12	0.24	−0.33	0.17	0.35	0.03
	C_0.01_	0.20	−0.05	0.07	0.32	0.04	0.44	0.21	0.02

## Data Availability

Unable to obtain data due to privacy concerns.

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
