# Peer review of "Study on the Key Autotoxic Substances of Alfalfa and Their Effects"

_plants, 2023, doi:10.3390/plants12183263_

Round 1
Reviewer 1 Report
The peer-reviewed article is characterized by integrity of presentation and good scientific reasoning. Many different methods have been used to support the authors' scientific hypothesis. In my opinion, the article lacks a practical result. In conclusion, I would like to see the author's recommendation on the possibility of cultivating this or that variety. Does the high content of coumarins in variety LZ affect the economic evaluation of the variety? Which of the varieties should be sown on a larger scale? If autotoxication is so high, is it better to sow alfalfa in grass mixtures? I think this kind of addition will lead to more readers' interest in this article.
Author Response
请参阅附件

Reviewer 2 Report
Manuscript ID: plants-2590292
Type: Article
Title: Study on Key Autotoxic Substance and Autotoxicity of alfalfa
Authors: Bei Wu, Shang Li Shi*, Huihui Zhang, Yuanyuan Du, Fang Jing
Recommendation: Minor Revision
Overall, the manuscript is good and very suitable for possible publication in the journal.
1. The title should be eye-catching and interesting. The title “Study on Key Autotoxic Substance and Autotoxicity of alfalfa”, here autotoxic and Autotoxicity are used two times. However, the title is very short, the authors are requested to change the same.
2. This article is written very well and may be helpful for readers around the globe.
3. The abstract is good and to the point.
4. The introduction section is written very well; however, it is very long.
5. The material and methods section has been described in detail. The hard work of the authors is highly appreciated.
6. Results and discussion are appropriately discussed. The overall manuscript is good. However, there are some errors that have been observed which need to be improved before publication of the article.
7. Tables and figures are nicely presented.
8. The Conclusion section is good. This section is very lengthy. I think it needs revision/modifications. It seems like the authors have pasted the contents from the abstracts section. I am sorry for these comments. However, the conclusion section needs refining before the final publication of this article.
Reviewer 3 Report
The presented manuscript is devoted to assessing the phytotoxicity of
various varieties of alfalfa. It is shown that this parameter is
determined by the varietal characteristics of the crop. The obtained
result can be used to create new highly productive varieties. These
varieties must have a high nitrogen-fixing ability, which is very
important for increasing the involvement of biological nitrogen in
agrocenoses. This can significantly reduce the use of nitrogen
fertilizers for crops. It is advisable to present these questions in the
article.
